# Validation and Psychometric Properties of the Spanish Version of the Fear of Childbirth Questionnaire (CFQ-e)

**DOI:** 10.3390/jcm11071843

**Published:** 2022-03-26

**Authors:** Héctor González-de la Torre, Adela Domínguez-Gil, Cintia Padrón-Brito, Carla Rosillo-Otero, Miriam Berenguer-Pérez, José Verdú-Soriano

**Affiliations:** 1Research Unit, Insular Maternal and Child University Hospital Complex of Gran Canaria, Canary Health Service, 35016 Las Palmas de Gran Canaria, Spain; 2Department of Nursing, Nursing School La Palma, University of La Laguna, 38200 San Cristóbal de La Laguna, Spain; 3Obstetrics and Gynaecology Department, Insular Maternal and Child University Hospital Complex of Gran Canaria, Canary Health Service, 35016 Las Palmas de Gran Canaria, Spain; adeadg96@gmail.com (A.D.-G.); cintiapb992@gmail.com (C.P.-B.); carlarosillo@gmail.com (C.R.-O.); 4Department of Community Nursing, Preventive Medicine, Public Health and History of Science, Faculty of Health Sciences, University of Alicante, 03690 Alicante, Spain; miriam.berenguer@ua.es

**Keywords:** fear of childbirth, pregnancy, surveys and questionnaires, validation studies as topic

## Abstract

The fear of childbirth is a topical concern, yet the issue has barely been studied in Spain, and only one fear of childbirth measurement instrument has been validated in the country. The aim of this study was to translate, adapt and validate the Fear of Childbirth Questionnaire (CFQ) for use in Spain, as well as to describe and evaluate the psychometric properties of the Spanish version of this instrument. In a first phase, a methodological study was carried out (translation–backtranslation and cross-cultural adaptation), and pilot study was carried out in the target population. In addition, content validation of the instrument was obtained (CFQ-e) from 10 experts. In the second phase, a cross-sectional study was carried out at several centres in Gran Canaria Island to obtain a validation sample. The evaluation of the psychometric properties of the CFQ-e, including construct validity through exploratory factor analysis and confirmatory factor analysis, the calculation of reliability via factor consistency using the ORION coefficients as well as alpha and omega coefficients were carried out. The CFQ-e showed evidence of content validity, adequate construct validity and reliability. The CFQ-e is composed of 37 items distributed in four subscales or dimensions: “fear of medical interventions”; “fear of harm and dying”; “fear of pain” and “fears relating to sexual aspects and embarrassment”. The CFQ-e constitutes a valid and reliable tool to measure the fear of childbirth in the Spanish pregnant population.

## 1. Introduction

The fear of childbirth (FOC) is a state of intense anxiety that leads some pregnant women to a fear of childbirth that interferes with their daily lives [1,2,3].

This fear can become pathological, in which case, it is called tokophobia and can negatively affect the development of the pregnancy and childbirth, as well as favour the development of post-traumatic stress disorders, postpartum depression and anxiety [4,5,6,7].

There seems to be a consensus that both FOC and tokophobia rates are increasing in pregnant women [3,8]. The global prevalence of FOC is, however, difficult to estimate [3,9,10]. A global FOC prevalence of 14% has been suggested [9], but this figure is disputed; reported rates vary widely from one study to another, and the problem is often under-detected [11,12].

There are various possible explanations for this. On the one hand, different definitions of the fear of childbirth and tokophobia have been proposed. This makes it very difficult to establish a clear divide between both conditions, distorting the calculations of real and accurate prevalence rates [3,8,13]. On the other hand, FOC is known to be linked to certain factors. For example, FOC prevalence differs according to the woman’s previous number of childbirths, showing higher rates in nulliparous women compared to multiparous women [14,15]. Some authors therefore argue that both groups of women should be studied separately regarding FOC [16,17]. 

Nevertheless, a decisive factor in the detection, diagnosis and evaluation of FOC is the availability of various measurement instruments [18,19]. The use of different scales or instruments conditions the calculation of cut-off points and the determination of mild, moderate or severe fear. It also obstructs the comparative analyses of different studies that address this issue [8,10,20].

Although many tools and measuring instruments exist to assess the fear of childbirth, the most widely known and applied is the Wijma Delivery Expectancy/Experience Questionnaire (W-DEQ) [10,18,19]. Developed in Sweden in 1998, the W-DEQ is a two-part questionnaire (W-DEQ-A and W-DEQ-B) [21]. Part A measures FOC based on women’s expectations, and Part B measures FOC based on prior experience [21]. 

The W-DEQ has been validated in several languages and settings and has been extensively used in FOC studies [10,18,20]. It is worth noting that the W-DEQ was initially conceived as a unidimensional tool [18], despite the causes of FOC being multifactorial [10,22,23].

Other multidimensional tools have thus been developed to measure FOC. One example is the Slade–Pais Expectations of Childbirth Scale (SPECS) [24], or more recently, the Fear of Childbirth Questionnaire (CFQ) [17,22,25].

The CFQ’s creators designed the instrument so that it could both measure FOC symptom severity and be used as a screening tool for clinically significant symptoms. To do this, the CFQ includes a wide range of fears related to childbirth that can be perceived by women, reflected in 40 items and organised into 9 dimensions or subscales [17,22,25]. This feature is important, as it allows one to detect and determine the domains in which health professionals should further educate and/or intervene when addressing a pregnant woman with FOC. Additionally, the CFQ includes another scale that measures the interference of FOC in the different spheres of the pregnant woman’s life [17].

An increased risk of elective caesarean section has been linked to severe cases of FOC [26,27]. The CFQ design therefore took into account the fact that it was useful for measuring the fear of both vaginal delivery and caesarean section.

The CFQ was validated in a sample of 643 pregnant women from different English-speaking countries (Canada, the United States and the United Kingdom). A Cronbach’s alpha of 0.94 was obtained for the general 40-item scale and 0.85 for the interference scale [17,25].

The fear of childbirth is widely studied in certain countries, such as Scandinavian countries [20]. In Finland, for example, FOC measurement and treatment is regulated during pregnancy [5], and in Sweden, midwives offer routine counselling [28]. In Spain, the existence of FOC has been recognised [29], but the problem has barely been studied, partly due to a lack of validated measurement tools [19]. The W-DEQ-A [30] and W-DEQ-B [31] have only very recently been validated in Spain. 

Given the growing interest in this topic and the lack of measurement instruments in Spain, the aim was to translate, adapt and validate the Fear of Childbirth Questionnaire (CFQ) for use in Spanish settings, as well as to describe and evaluate the psychometric properties of the Spanish version of this instrument.

## 2. Materials and Methods

The present study took place over two phases: 

First Phase: The translation and cross-cultural adaptation of the CFQ and content validation by experts.

Second Phase: A cross-sectional observational study to evaluate the psychometric properties of the Spanish version of the CFQ (CFQ-e).

### 2.1. Phase 1

#### 2.1.1. Starting Instrument

The original version of the CFQ consists of 40 items based on a positive Likert scale ranging from 0 points to 4 points. The total score can therefore range from 0 to 160 points (the higher the score, the greater the fear). These items are grouped into 9 subscales, which represent different dimensions or constructs of the fear of childbirth [17,25].

The subscales considered were as follows: fear of loss of sexual pleasure/attractiveness (6 items); fear of pain from a vaginal birth (5 items); fear of medical interventions (7 items); fear of embarrassment (5 items); fear of harm to baby (3 items); fear of caesarean birth (3 items); fear of mum or baby dying (3 items); fear of insufficient pain medication (3 items); fear of body damage from a vaginal birth (5 items). To obtain each subscale’s score, the scores of the items in each subscale are added up, and the sum was divided by the number of items within each subscale. This made it possible to compare the different subscale scores [17,22,25].

In addition, the CFQ includes another scale that measures the degree of FOC interference in the different spheres of the pregnant woman’s life. This scale consists of 7 Likert-type items, with 0 points meaning no interference and 4 points signifying extreme interference (between 0 and 28 points/the higher the score, the greater the interference) [17].

#### 2.1.2. Translation and Cross-Cultural Adaptation 

In order to translate and culturally adapt the questionnaire, the stages proposed by Sousa et al. were followed [32]. The author of the original questionnaire was first contacted to ask for her approval, and her authorisation to adapt it was obtained.

The process unfolded from April to June 2020. Two freelance translators performed two translations from English to Spanish. The first translator was a midwife who had completed her professional studies in England, was a native Spanish speaker, but had been bilingual English/Spanish since childhood. The second was Spanish and a professional translator. The first translator was knowledgeable in the subject, while the second was not. Both translations were analysed and discussed by the research team to obtain an initial unified version (preliminary version 1 of the CFQ-e).

This preliminary version 1 was sent to two independent bilingual translators—who were different from the initial translators—and they performed two backtranslations. The first was a native English translator, who was bilingual and had been residing in Spain for many years. The second was a native Spanish speaker, an obstetrician, who was also bilingual English/Spanish, having spent all her childhood in an English-speaking country (Australia). This second translator was knowledgeable in the subject, while the first was not.

The research team compared the two backtranslations with the original version of the questionnaire, discussing possible discrepancies until a consensus was reached. They were also sent to the original author via email for evaluation. No item was considered necessary to modify, and the similarity of the two backtranslations with the original version of the CFQ was confirmed. This phase thus led to preliminary version 2 of the CFQ-e.

Finally, this version was evaluated and compared with the original CFQ by an external bilingual researcher, an expert in Health Science research methodology and highly experienced in the adaptation and validation of questionnaires.

#### 2.1.3. Pretest

A pilot of the preliminary CFQ-e version 2 was administered to the population under study by means of convenience sampling in order to estimate the instrument’s feasibility and viability, as well as its cultural adequacy in the Spanish population. In this phase, the aim was to identify ambiguous items, possible errors and misunderstandings of the items, as well as to assess the burden of administering the questionnaire. Sampling was considered completed when none of the participants expressed any comprehension problems with the questionnaire.

#### 2.1.4. Content Validation

To evaluate the total content validity (CVI-T), the expert test described by Polit and Hungler was performed [33]. The content validity of each item (CVI-I) and the content validity index per expert (CVI-E), as well as the total content validity (CVI-T = sum of the CVI of each expert/total number of experts) were also calculated based on the expert scores. To ensure the validity of the items in the content validity index calculation, the likely random agreement (Pa) was corrected using the formula Pa = [N!/(A!(NA)!)] × 0.5N, where N = expert number and A = nº according to good relevance and the statistical calculation of the modified Kappa (K* = (CVI-I − Pa)/(1 − Pa)) for each instrument item [33,34].

### 2.2. Phase 2 

#### 2.2.1. Design

A cross-sectional observational study was proposed to obtain a validation sample for the CFQ-e questionnaire.

#### 2.2.2. Population to Study

The population to be studied was pregnant women living in Gran Canaria Island. The inclusion criteria were as follows: pregnant women aged 18 years or above, with a gestational age equal to or over 16 weeks and with pregnancies with a live foetus. The exclusion criteria were the following: scheduled/elective caesarean section, in active stage of labour and having a language barrier (difficulty reading or understanding the Spanish language).

Withdrawal criteria included incorrectly or incompletely completing the questionnaire (unanswered items or multiple answers where inappropriate) or wishing to leave the study after having given informed consent.

#### 2.2.3. Sampling and Data Collection

Non-probabilistic convenience sampling was applied. Participants were recruited among women whose pregnancy was being followed up at primary care centres and specialised care centres, or pregnant women who went to consultations and the emergency service of the Insular Maternal and Child University Hospital Complex of Gran Canaria. Each centre’s responsible obstetrician or midwife collaborated in the recruitment process. Data were collected from 1 August to 15 November 2020.

It is usually considered that at least 10 subjects need to be studied for each questionnaire item in order to obtain a sufficient number of subjects for an exploratory factor analysis (EFA) [35,36]. Given that the original questionnaire consisted of 40 items, the intention was to reach a minimum sample size of 400 pregnant women.

#### 2.2.4. Variables and Collection Instrument

The questionnaire consisted of two parts. The first part was created “ad hoc” to collect some sociodemographic variables (age, education level and marital status) and obstetric variables (gestational age, previous offspring, type of previous childbirths, how the current pregnancy was achieved, single or twin gestation and the existence of any risk factors). All these variables were gathered by the midwives or obstetricians from the clinical history records. 

The second part collected answers to the first version of the CFQ-e. It maintained the same number of items as the original CFQ (40 items) and the interference scale (7 items).

#### 2.2.5. Data Analysis

A descriptive analysis of the variables was conducted using the IBM© SPSS Statistics v.28.0 statistical program, expressing the qualitative variables in percentages and frequencies, and in the case of quantitative variables, in means, standard deviation and minimum-maximum values.

#### 2.2.6. Construct Validation

To evaluate construct validity, a factor analysis was performed using the FACTOR program v.11.05.01 [37,38,39]. To estimate whether the common variance justified a factor analysis, the Kaiser–Meyer–Olkin index (KMO) was used, with values above 0.75 being considered adequate, as well as Bartlett’s statistic, with values *p* ≤ 0.05 being considered statistically significant [35,36]. The following indices were used to evaluate the adequacy of the Factorial Solution: Root Mean Square Error of Approximation (RMSEA), Non-Normed Fit Index (NNFI), Comparative Fit Index (CFI), Goodness of Fit Index (GFI) and the Adjusted Goodness of Fit Index (AGFI). RMSEA values less than 0.05 were considered a good fit and values between 0.05 and 0.08 a reasonable fit [35]. NNFI and CFI values of 0.95 or higher were accepted as indicators of good fit, with values for GFI and AGFI over 0.90 generally indicating acceptable model fit [35,38].

To examine the questionnaire’s factorial structure, a random sample was initially selected using the Solomon method [35,40], based on 279 participants out of the 557 included in the total sample. An EFA was carried out with the Pearson correlation matrix (according to the result of the Mardia test for symmetry and kurtosis) and the extraction of factors by unweighted least squares and PROMIN rotation [35,41,42]. A parallel analysis was used to establish the number of factors to retain. The consistency (reliability) of the retained factors was calculated. Using bootstrapping, 95% confidence intervals (95%CI) were calculated for the model measurements. Initially, the EFA was performed for a nine-factor model to verify its similarity with the original model proposed for the CFQ, and, subsequently, for a four-factor model, according to the solution suggested in the first EFA, via parallel analysis.

Subsequently, a confirmatory factor analysis (CFA) was performed using the data of the remaining 278 participants, taking as a reference the factor loadings matrix obtained from the first sample’s EFA. The loading matrix was semi-specified, with non-zero values attributed to coefficients above 0.30 for each factor and zero to the rest. In cases in which an item’s factor loading was above 0.30 in more than one factor, a value other than 0 was assigned in the one with the highest loading and 0 in the rest.

#### 2.2.7. Internal Consistency

Factor consistency was evaluated using the ORION coefficients (Overall Reliability of fully Informative prior Oblique N-EAP scores) [35,43]. Moreover, the questionnaire’s reliability was calculated based on the alpha coefficient and the omega coefficient using the IBM Corp. Released 2021. IBM SPSS Statistics for Windows, Version 28.0. Armonk, NY, USA: IBM Corp. 

#### 2.2.8. Ethical Considerations

The study protocol was evaluated and approved by the Research Ethics Committee/Medicines Research Ethics Committee Universitary Hospital of Gran Canaria Dr. Negrín (CEI/CEIm HUGCDN, CEIm HUGCDN Code: 2020-264-1). The project was explained to each participant, and their written informed consent was obtained. An anonymised matrix was used to statistically analyse the data. Confidentiality and anonymity were ensured during all of the stages of the study.

## 3. Results

### 3.1. Phase 1 

#### 3.1.1. Translation and Cross-Cultural Adaptation 

The research team’s review of the translated versions revealed no major discrepancies between the two initial translations. No items were controversial, and preliminary version 1 CFQ-e was obtained.

The research team compared the original questionnaire with the two backtranslations obtained from CFQ-e preliminary version 1. No significant differences were found between the two versions. The original author also evaluated them and considered that both backtranslations were faithful and conveyed the meaning of the original questionnaire. The author did show, however, a slight preference for backtranslation number 2. 

The external researcher agreed that CFQ-e preliminary version 2 was similar to the CFQ original version.

#### 3.1.2. Pretest

A total of 20 pregnant women presenting similar inclusion and exclusion criteria to those considered in phase 2 answered and completed CFQ-e preliminary version 2 in writing, expressing their opinion and providing suggestions. This pilot study was conducted in one of the health centres participating in the study and in the emergency department of the Insular Maternal and Child University Hospital Complex of Gran Canaria.

Overall, no problems were encountered regarding the participants’ understanding of the questionnaire, except for item number 19 (“having an episiotomy”), since the pretest revealed that several women did not understand the term “episiotomy”. This meant that the item had to be modified with an additional clarification note, as follows: Item 19—have an episiotomy performed (have a cut made in your vagina). This item was considered to be the one that differed the most from the original CFQ reference item. Regarding the efforts required to fill out the questionnaire, a number of pregnant women found that it was rather long to complete.

After the pretest, the first Spanish version of the CFQ (CFQ-e) was obtained. Similarly to the original CFQ, it included 40 items (Appendix A: First version of the CFQ-e).

#### 3.1.3. Content Validation

The expert panel was composed of ten professionals: six women and four men (two obstetricians, five midwives and three nurses). The expert assessment provided a CVI-Total of 0.77. The CVI-E values ranged from 0.52 (one expert) to 1 (two experts) (Appendix A: Profile and CVI-E of the experts). According to the experts, twenty-one out of the forty items included in the CFQ-e showed an excellent CVI-I, and another ten items obtained good CVI-I. Nine items presented a CVI-I with “fair” (items 14, 21, 24 and 32) or “poor” (items 5, 12, 13, 22 and 38) values (Appendix A: CVI-I scores for each CFQ-e item).

### 3.2. Phase 2

#### 3.2.1. Sociodemographic Characteristics 

A total of 608 women from 22 health centres, 3 specialised care centres, obstetric consultations and the emergency service of the XXXX completed the questionnaires, but 51 questionnaires were inadequately completed and had to be withdrawn. The final sample was therefore composed of a total of 557 women (*n* = 557).

The participants’ mean age was 31.30 years (SD = 5.49/Minimum = 18-Maximum = 48). The mean gestational age was 29.63 weeks (SD = 7.42/Minimum = 16.00-Maximum = 42.00). Table 1 shows the frequencies and percentages for the rest of the sociodemographic and obstetric variables considered.

The final mean score obtained for fear of childbirth in the sample was 66.15 points (SD = 26.77/Minimum = 1.00-Maximum = 143.00). The final mean score obtained for the interference scale in the sample was 4.95 points (SD = 5.06/Minimum = 0-Maximum = 28). Table 2 shows the floor percentage, ceiling percentage, mean and standard deviation for each item, as well as the average scores for the nine subscales considered in the original CFQ. These same values can be found for the interference scale in Appendix A.

#### 3.2.2. Preliminary Factor Analysis

Initially, a preliminary EFA was performed for a nine-factor model, according to the model proposed in the original questionnaire. Although it presented very good adequacy, with a Kaiser–Meyer–Olkin measure (KMO) = 0.918 (95%CI: 0.867–0.911) and a significant Bartlett statistic (*p* = 0.00001), and it showed excellent goodness of fit indices (Root Mean Square Error of Approximation (RMSEA) = 0.000 (95%CI: could not be computed), Non-Normed Fit Index (NNFI) = 1.020 (95%CI: 1.015–1.026) and Comparative Fit Index (CFI) = 0.999 (95%CI: 0.999–0.999), the parallel analysis recommended a four-factor solution. Because of this, an EFA was carried out for a four-factor model.

The EFA for this four-factor model presented very good adequacy, with KMO = 0.918 (95%CI: 0.867–0.911) and Bartlett’s statistic *p* = 0.00001, with the goodness of fit indices being RMSEA = 0.021 (95%CI: 0.015–0.015), NNFI = 0.996 (95%CI: 0.997–0.998) and CFI = 0.997 (95%CI: 0.997–0.999). In this model, all items had a factor loading above 0.30 in the assigned factor, except item numbers 8 and 14, which did not obtain sufficiently satisfactory loading for any factor (less than 0.30 in both cases) (Appendix A: Factor loadings obtained in the EFA of the model of 4 factors and 40 items).

The CFA subsequently performed on the second sample (278 participants) to confirm the model showed excellent adjustment (RMSEA = 0.022 (95%CI: 0.001–0.022), NNFI = 0.996 (95%CI: 0.995–1.000) and CFI = 0.997 (95%CI: 0.996–1.000) but suggested problems in the case of some items. Thus, two items (items 11 and 14) had loadings below 0.30. Moreover, a change of factor assignment with respect to the EFA was proposed for another, with the factor loading of “Item 19-have an episiotomy performed (have a cut made in your vagina)” being the lowest of all those that exceeded the established limit of 0.30 (0.311) (Appendix A: Factor loadings obtained in the CFA of the four-factor forty-item model). The analysis of the estimated congruences for each item [44] indicated comparatively lower values for these three items compared to the rest, especially in the case of item 19 (Congruence Index = 0.208/95%CI: −0.165–0.531) (Appendix A).

Based on the results obtained in this preliminary factor analysis, a CFQ-e four-factor model with thirty-seven items was tested via a new factor analysis, eliminating items number 11, 14 and 19 of the first version of the questionnaire and maintaining the rest of the conditions that were specified in the Materials and Methods section.

#### 3.2.3. Exploratory Factor Analysis (EFA) of the Four-Factor and Thirty-Seven-Item Model

The EFA for this model also presented very good adequacy, with KMO = 0.913 (95%CI: 0.864–0.908) and a significant Bartlett statistic (*p* = 0.00001), with Normed item-MSA indices above 0.85 in all items. The four-factor solution provided a total explained variance of 58.75% according to the parallel analysis. The goodness of fit indices for the model were RMSEA = 0.025 (95%CI: 0.013–0.024)—i.e., below the 0.05 limit to be considered a good fit—NNFI = 0.995 (95%CI: 0.994–0.999), CFI = 0.996 (95%CI: 0.995–0.999) and above 0.95, also indicating an excellent fit.

Table 3 presents the factor loadings (after rotation) of the four-factor and thirty-seven-item model. According to the analysis, in this model, Factor 1 groups all the “fear of caesarean birth“ and “fear of medical interventions” subscale items (except items 1 and 25) in addition to item 28; Factor 2 includes all the items relating to “fear of harm to baby”, “fear of mum or baby dying” and “fear of body damage from a vaginal birth” subscales, plus item 1; Factor 3 includes all the items from the “fear of pain from a vaginal birth” and “fear of insufficient pain medication” subscales, and finally Factor 4, includes the items from the subscales “fear of loss of pleasure/sexual attractiveness” and “fear of embarrassment”, plus item 25.

In this initial model, F1 is called “fear of medical interventions” (9 items), F2 is “fear of harm and dying” (10 items), F3 refers to “fear of pain” (7 items) and Factor 4 is called “fear relating to sexual aspects and embarrassment” (11 items).

The results obtained can generally be observed to maintain the structure of the original questionnaire, although some subscales were grouped until they reached a reduction in four dimensions or subscales. With this structure, all items had a factor loading above 0.30 in the assigned factor, with the lowest value corresponding to item number 8 (0.307). The items could have been assigned differently to the factors, since some items (items number 18, 23 and 25) had a loading above 0.3 in several factors. These items could therefore be assigned to the factor presenting the highest factor loading to perform the subsequent CFA.

#### 3.2.4. Confirmatory Factor Analysis (CFA) for the Four-Factor and Thirty-Seven-Item Model

The factorial model obtained from four factors with the first sample (*n* = 279) was confirmed via CFA using the second sample with the remaining 278 participants. To do this, a CFA was performed using a semi-specified matrix of factor loading coefficients. This procedure compares the congruence or similarity with a model for which the factor loadings are 0 in specified items and other than 0 in the rest. Accordingly, the factor loadings matrix to be confirmed were factor loadings other than 0 in the items obtained from the EFA of the four-factor and thirty-seven-item model in the first sample. 

The second sample (*n* = 278) presented very good adequacy, with a KMO = 0.914 (95%CI: 0.864–0.908) and a significant Bartlett statistic (*p* = 0.00001), and with an explained variance by the four factors of 59.84% according to the parallel analysis. The model’s goodness of fit indices were RMSEA = 0.028 (95%CI: 0.018–0.030) (below the 0.05 limit to be considered a good fit), NNFI = 0.994 (95%CI: 0.992–0.998) and CFI = 0.995 (95%CI: 0.993–0.998), above 0.95, thus confirming the model’s excellent fit. The Goodness of Fit Index (GFI) was 0.984 (95%CI: 0.981–0.988), and the Adjusted Goodness of Fit Index (AGFI) = 0.980 (95%CI: 0.976–0.984).

Table 4 presents the model’s factor loadings (after rotation) obtained after the CFA. All items had loadings above 0.30. The analysis confirmed the assignment of most items to the factors proposed by the EFA, although three items had loadings above 0.30 in two factors (items number 18, 25 and 28). Depending on the loadings, the CFA led to a change in factor assignment for items 18 and 25. 

“Item 18—not receiving adequate pain relief” had loadings for factors F2 (0.470) and F3 (0.386). Although the highest loading was for F2, it was considered more appropriate and in line with the theoretical framework to maintain this item in factor F3 (“fear of pain”).

Regarding “item 28—needing stitches after childbirth”, despite receiving loadings for two factors (F1 and F4), the F4 loading was minimal (0.309), so the factor change was not considered and F1 was maintained, as suggested by the EFA. 

The only item that was considered problematic in this model was “item 25—scars after a C-section”, since it had very similar loadings for Factors F1 (0.388) and F4 (0.376). In this case, a change of factor was chosen, assigning it to F1 (fear of medical interventions).

Appendix A illustrates the root mean square discrepancy (RMSD) between the rotated loading matrix and the target matrix for each variable under study between the data of the second sample and the semi-specified four-factor model. The global RSMDs estimated for each factor were 0.086 (95%CI: 0.065–0.100) for F1, 0.108 (95%CI: 0.077–0.128) for F2, 0.104 (95%CI: 0.078–0.115) for F3 and 0.120 (95%CI: 0.085–0.137), with an overall model discrepancy coefficient of 0.105 (95%CI: 0.103–0.104). Table 5 shows the correlations between the model factors, presenting all significant factor correlations.

#### 3.2.5. CFQ-e Final Instrument

The analysis gave rise to a final version of the CFQ-e, consisting of 37 items (items number 11, 14 and 19 in the original CFQ questionnaire were eliminated), distributed in 4 dimensions or subscales called: “fear of medical interventions” (10 items), “fear of harm and dying” (10 items), “fear of pain” (7 items) and “fear relating to sexual aspects and embarrassment” (10 items). In this way, the total score can range between 0 and 148 points (a higher score corresponds to greater fear). The interference scale remains unchanged. The final version of the CFQ-e can be found in Appendix A.

#### 3.2.6. Internal Consistency

The values obtained for the ORION coefficients for the final version of the CFQ-e were 0.900 (95%CI: 0.800–0.916) for F1, 0.954 (95%CI: 0.938–0.962) for F2, 0.940 (95%CI: 0.924–0.955) for F3 and 0.930 (95%CI: 0.914–0.941) for F4. All values were above 0.80, thus showing adequate consistency [43,45].

A total alpha of 0.947 was obtained for the final version of the CFQ-e and 0.898 for the interference scale. The total omega coefficient was 0.945 for the CFQ-e and 0.898 for the interference scale. All alpha and omega coefficient values calculated for the subscales in this study can be seen in Appendix A.

## 4. Discussion

Despite extensive interest in FOC and the large amount of research published on the subject, FOC has barely been studied in Spain [30,46]. The absence of validated FOC measurement tools in Spain has undoubtedly contributed to this situation.

Some authors have pointed out the need for measuring instruments that assess the different dimensions that could be related to the fear of childbirth [8]. The FCQ’s multidimensionality is regarded by its creators as one of its greatest strengths compared to the W-DEQ [17,25], which was hitherto considered the “gold standard” of FOC measurement [10,18,30].

The unidimensional nature of the W-DEQ has been much discussed, as many studies suggest that this measuring instrument should in fact be considered multidimensional [47,48,49]. Despite this, no consensus has been reached regarding the number of factors or dimensions included or their composition [50]. In fact, the authors who performed the Spanish validation (W-DEQ-A-Sp) suggested that this version had four factors or dimensions [30].

Unlike W-DEQ, the CFQ was conceived as a multidimensional tool from the outset. For this reason—and because another research group was already validating the W-DEQ in Spain—we chose the CFQ as the tool to validate, regarding it as a potentially effective way to study FOC in the Spanish population.

The performed factor analysis indicates that the Spanish version of the CFQ-e is composed of thirty-seven items and four factors/subscales, achieving an appropriate model adjustment. The model’s goodness of fit indices were greater than that of the original version of the CFQ, with RMSEA = 0.028 (95%CI: 0.018–0.030) and CFI = 0.995 (95%CI: 0.993–0.998) compared to RMSEA = 0.064 (90% CI: 0.062–0.066) and CFI = 0.977, reported by Fairbrother [25].

We cannot yet compare these results with other validation studies in other language populations since, to the best of our knowledge, this is the first CFQ validation study in a non-English-speaking setting. Further validation studies are required in other contexts to provide more information in this regard and to confirm or discard this model.

Regarding the sample size, we followed the classic factorial analysis recommendation of having at least 10 subjects for each item. We do accept, however, that such a recommendation is highly controversial [36]. Conversely, when using Pearson correlation matrices, as in this case, the recommendation is to use a minimum sample of 200 subjects [35,36]. We therefore consider that the sample size achieved was sufficient to ensure the results’ internal validity. The KMO and Barlett statistic values obtained supported this assumption. 

While the EFA was initially conducted for nine subscales and showed excellent goodness of fit indices, the parallel analysis suggested four factors. A parallel analysis method was chosen, since this system allows one to perform the most rigorous identification of a questionnaire’s numbers of dimensions [36,51]. In addition, it was also used by Fairbrother et al. and Ortega-Cejas et al. to validate the W-DEQ-Sp [25,30,31].

The number of items that should be included in each factor is an object of discussion. The common procedure is to select a minimum of three items with high saturations (factor loadings above 0.60) by factor [36,52]. This practice, however, has been described as counter-productive, as it can affect the stability of the results [36]. It seems clear that the greater the number of items, and the more accurately they measure a factor, the more stable the factor solution [35,36]. The distribution of the items and factor loadings obtained for the four-factor model proposed for CFQ-e is robust in this regard.

The analysis revealed problems with three items. Item numbers “11—that you do not have a caesarean section when it is what you want” and “14—losing control of your emotions in front of other people (being rude, screaming) during childbirth” had loadings below 0.30 in the preliminary factor analysis. These items obtained kappa values of 0.66 (good) and 0.50 (fair), respectively, in the content validation process. The latter suggests that the experts had assessed, a priori, the existence of cultural aspects that could affect the adequacy of these items in Spanish settings. 

In Fairbrother’s factor analysis, the item “11—that you do not have a caesarean section when it is what you want” was the item with the lowest factor loading in the EFA (0.297), discarding items with similar and even higher loadings during the creation and validation of the CFQ [25]. These results indicate that this item is problematic in both versions, perhaps because there may be a contradiction between women who desire a caesarean section and those who wish to avoid it at all costs, which is reflected in the item’s construct. 

Fairbrother’s study did not identify any problems with the item “14—losing control of your emotions in front of other people (being rude, screaming) during childbirth”, with loadings above 0.35 (0.446), although this item had the lowest loading in its dimension [25]. The results of our study indicate that the item is complex, at least in the case of our sample, since it had much lower loadings, as well as the highest loading (0.208) in a subscale factor which was not very consistent with the theoretical framework (F4–fear of harm and dying). Although a possible explanation may be that Spanish women are not afraid of losing control of their emotions during labour, this item’s average score in our sample was 0.94, i.e., higher than that obtained for other items of the original subscale. The relationship between the loss of self-control during labour and fear in the Spanish population should thus be explored more in depth.

“Item 19–Have an episiotomy performed (have a cut made in your vagina)” deserves special attention. In the pilot study, it was found that several women did not understand this term, so it had to be substantially modified with respect to that of the original questionnaire. Although the item obtained a discrete factor loading (0.311), the estimated congruence value was extremely low (Congruence Index = 0.208). While this index has given rise to different interpretations, most authors agree that values above 0.85 can be considered adequate, indicating a similarity of the items with the model [44]. Values below 0.68 are considered “terrible” [44]. Based on the latter, item 19 was removed from the final CFQ-e model, although we recommend assessing this aspect in future studies on the CFQ-e. In Spain, episiotomy rates are even higher than the recommended number [53,54,55], suggesting that Spanish women are still insufficiently aware of certain childbirth interventions, and this could influence FOC.

The obtained alpha coefficients, with a total value of 0.947, and values above 0.85 for the four factor-subscales indicated the adequate reliability of the final version of the CFQ-e for practical use [56], the values being almost identical to those reported by Fairbrother and her team [17,25].

In recent years, the widespread use of the alpha coefficient as the only index to evaluate the reliability of a measuring instrument has been criticised [57,58]. This has led some authors to recommend the use of other estimators such as the omega coefficient [59,60]. Indeed, this latter index, unlike the alpha coefficient, works directly with the factor loadings, making the calculations more stable and reducing the dependence on the number of instrument items to be evaluated [58,61]. Values above 0.80 are considered adequate [59,61].

This aspect was not taken into account by the authors of the CFQ, who did not report the omega coefficients. They were, however, considered by those responsible for validating the W-DEQ-A-Sp, who mentioned a total omega coefficient of 0.936 and values between 0.80 and 0.90 for the four identified factors [30].

In our study, the omega coefficients were calculated both for the original nine-subscale model proposed by the original CFQ and for the final model of the proposed four-subscale and thirty-seven-item CFQ-e. The analysis indicated that despite an almost complete absence of variation in the total omega coefficient in our sample, the subscale values obtained were more suitable for the four-factor model of the CFQ-e. Indeed, the values were above 0.85 for all the subscales, while in the original model, the omega values were below 0.80 in three subscales.

Based on these results, we can affirm that the decision to remove the items mentioned above did not affect the reliability of the CFQ-e. Considering that, as identified in the pilot study, some women found that the questionnaire was long to complete, this decision improves the instrument’s applicability in practice.

Fairbrother et al., in their assessment of the CFQ’s convergent validity, reported a correlation value between CFQ and W-DEQ-A of 0.58 (*p* < 0.001) [25]. Since both the validation process of the W-DEQ-A-Sp and this study were conducted during the same period, it was not possible to explore the convergent validity between both tools. This step has thus been left for future studies.

Another unresolved question is the CFQ-e cut-off points between moderate fear and extreme fear, compared to the 83 and 104 points, respectively, proposed for the CFQ [17]. Further studies in the rest of Spain should explore and confirm the cut-off points, taking into account the fact that the scores must be adjusted (total score between 0 and 148 points for the CFQ-e, unlike the CFQ, whose total score can range between 0 and 160 points). Another line to explore is converting the scores to a standard scale, for example, with values 0–100, which would facilitate comparisons with other studies.

The study presented a number of limitations. We chose non-probabilistic convenience sampling, i.e., a type of sampling which presents certain shortcomings. We do not believe, however, that this affected the sample’s representativeness. Indeed, pregnant women were recruited from a large number of health centres. Moreover, numerous professionals, both obstetricians and midwives, also participated in the process. In addition, the sociodemographic and clinical characteristics of the Canarian’s pregnant population can be considered to be similar to the rest of Spain’s pregnant women, so the CFQ-e properties can be generalised.

A notable limitation is that women were recruited exclusively from public health services. Therefore, no data were obtained for women who had chosen to follow-up their pregnancies exclusively in private centres or services.

The present study was conducted in 2020, the year in which the SARS-CoV-2 global pandemic unfolded, also affecting pregnant women around the world [62]. This situation had an impact both on the data collection and women’s emotions, so it should be considered as a possible external confounding factor. In this respect, several studies have highlighted the influence of the pandemic on pregnant women’s mental health [63,64,65], so we can consider that the pandemic did influence the results obtained in this study.

## 5. Conclusions

The present work is the second validation study of a FOC evaluation instrument in Spain and the first validation of the CFQ in a non-English-speaking setting. The Spanish version of the CFQ (CFQ-e) consists of thirty-seven items distributed into four subscales or dimensions: “fear of medical interventions”, “fear of harm and dying”, “fear of pain” and “fears relating to sexual aspects and embarrassment”. The psychometric characteristics of the CFQ-e indicate that this instrument is useful, valid and reliable to measure fear of childbirth. In addition, it allows one to assess different dimensions associated with FOC in the Spanish population.

Further studies are needed to evaluate the prevalence of FOC in Spanish settings, as well as to explore the convergent validity of CFQ-e with other FOC measurement instruments.

## Figures and Tables

**Table 1 jcm-11-01843-t001:** Sociodemographic and obstetric variables of the sample (*n* = 557).

Variables	Frequency (%) *n* = 557	M (SD)
Age (years)		31.30 (5.49)
Gestational Age (weeks)		29.63 (7.42)
Level of studies		
No studies	2 (0.4)	
Primary education	127 (22.8)	
Secondary education	229 (41.1)	
University studies	199 (35.7)	
Marital status		
Has a partner	536 (96.2)	
No Partner	21 (3.8)	
Type of Pregnancy		
Single pregnancy	550 (98.7)	
Twin pregnancy	7 (1.3)	
How the current pregnancy was achieved		
Spontaneous	529 (95.0)	
Assisted reproduction technique	28 (5.0)	
Previous offspring ^a^		
Nulliparous	365 (65.5)	
Primiparous	146 (26.2)	
Multiparous	46 (8.3)	
Existence of at least one risk factor ^b^		
Yes	99 (17.8)	
No	458 (82.2)	
Gestational hypertension risk factor		
Yes	24 (4.3)	
No	533 (95.7)	
Preeclampsia risk factor		
Yes	4 (0.7)	
No	553 (99.3)	
Pregestational Diabetes risk factor		
Yes	4 (0.7)	
No	553 (99.3)	
Gestational Diabetes risk factor		
Yes	60 (10.8)	
No	497 (89.2)	
Intrauterine Growth Restriction risk factor		
Yes	9 (1.6)	
No	548 (98.4)	
Coagulopathies risk factor		
Yes	7 (1.3)	
No	550 (98.7)	
Anterior Eutocic delivery		
Yes	173 (31.1)	
No	384 (68.9)	
Anterior Dystopian delivery (Forceps)		
Yes	20 (3.6)	
No	537 (96.4)	
Previous caesarean section		
Yes	31 (5.6)	
No	526 (94.4)	

M = Mean/SD = standard deviation; ^a^ = nulliparous: woman with no vaginal births/primiparous: woman who had given birth vaginally only once/multiparous: woman who had had two or more vaginal births; ^b^ = existence of at least one of the risk factors considered in the study.

**Table 2 jcm-11-01843-t002:** Floor and ceiling scores and means and standard deviations for each of the items in the CFQ-e (*n* = 557).

Subscales and Items of the CFQ ^a^	Floor Not at All ^b^ *n* (%)	Ceiling Extremely ^b^ *n* (%)	M (SD)
Subscale Fear of loss of sexual pleasure/attractiveness			1.07 (0.86)
12—That your vagina stretches by having a vaginal birth	272 (48.8%)	9 (1.6%)	0.86 (1.02)
13—Enjoy less sexual intercourse by stretching the vagina because of vaginal birth	212 (38.1%)	19 (3.4%)	1.17 (1.15)
15—That your body is less attractive after childbirth	254 (45.6%)	18 (3.2%)	0.90 (1.05)
24—Make your vagina look less attractive after a vaginal birth	301 (54.0%)	9 (1.6%)	0.73 (0.94)
26—Enjoy less sexual intercourse because you feel pain or discomfort after childbirth	127 (22.8%)	25 (4.5%)	1.47 (1.15)
27—That your partner enjoys less of sexual intercourse after childbirth by stretching your vagina after childbirth	175 (31.4%)	20 (3.6%)	1.30 (1.15)
Subscale Fear of pain from a vaginal birth			1.33 (0.95)
30—Feeling pain during childbirth	117 (21.0%)	59 (10.6%)	1.77 (1.28)
31—Have a vaginal birth	333 (59.8%)	9 (1.6%)	0.71 (1.02)
34—Feel pain while pushing the baby	139 (25.0%)	20 (3.6%)	1.35 (1.09)
35—Feeling pain during a vaginal birth	133 (23.9%)	27 (4.8%)	1.41 (1.12)
37—Feeling pain during contractions	120 (21.5%)	22 (3.9%)	1.45 (1.08)
Subscale Fear of medical interventions			1.45 (0.79)
1—That you are harmed by incompetent medical assistance	44 (7.9%)	104 (18.7%)	2.25 (1.22)
4—Receive general anaesthesia	162 (29.1%)	38 (6.8%)	1.44 (1.24)
5—To be given injections	295 (53.0%)	15 (2.7%)	0.85 (1.09)
22—That you get the epidural	252 (45.2%)	27 (4.8%)	0.99 (1.15)
25—That you have scars after the caesarean section	288 (51.7%)	14 (2.5%)	0.82 (1.03)
38—To be probed (a tube that is inserted into the urethra to collect urine)	136 (24.4%)	57 (10.2%)	1.70 (1.31)
39—Feeling pain during a C-section	63 (11.3%)	88 (15.8%)	2.15 (1.24)
Subscale Fear of embarrassment			0.72 (0.69)
7—That other people see you naked during childbirth	428 (76.8%)	2 (0.4%)	0.34 (0.70)
14—Losing control of your emotions in front of other people (being rude, screaming) during childbirth	241 (43.3%)	14 (2.5%)	0.94 (1.04)
21—That other people see you urinating during childbirth	328 (58.9%)	7 (1.3%)	0.63 (0.90)
23—Feeling observed by strangers during childbirth	355 (63.7%)	8 (1.4%)	0.58 (0.91)
32—That other people see you defecate during childbirth	216 (38.8%)	26 (4.7%)	1.12 (1.17)
Subscale Fear of harm to baby			3.25 (0.97)
6—That damage or harm the baby as a result of childbirth	19 (3.4%)	307 (55.1%)	3.18 (1.11)
9—That the baby suffers some damage during childbirth	9 (1.6%)	329 (59.1%)	3.33 (0.96)
10—That harm the baby in a medical intervention during childbirth (e.g., vacuum, anaesthesia, forceps...)	13 (2.3%)	307 (55.1%)	3.24 (1.02)
Subscale Fear of caesarean birth			1.72 (1.10)
33—Not being able to have a vaginal birth despite being what you prefer	132 (23.7%)	44 (7.9%)	1.58 (1.23)
36—Having a caesarean section	116 (20.8%)	73 (13.1%)	1.86 (1.31)
40—Not being able to have the type of birth you would like (for example, vaginal or caesarean section)	119 (21.4%)	63 (11.3%)	1.73 (1.28)
Subscale Fear of mom or baby dying			3.06 (1.06)
3—Dying during childbirth	98 (17.6%)	233 (41.8%)	2.46 (1.57)
16—That the baby suffocates during childbirth	15 (2.7%)	324 (58.2%)	3.24 (1.08)
20—That the baby dies during childbirth	18 (3.2%)	415 (74.5%)	3.49 (1.04)
Subscale Fear of insufficient pain medication			1.66 (1.01)
11—That you do not have a caesarean section when it is what you want	242 (43.4%)	32 (5.7%)	1.21 (1.29)
18—Not getting the pain medication you need	67 (12.0%)	57 (10.2%)	1.92 (1.18)
29—That you do not put the epidural during childbirth in the case of wanting it or needing it	102 (18.3%)	65 (11.7%)	1.87 (1.29)
Subscale Fear of body damage from a vaginal birth			2.01 (0.90)
2—Suffer a tear or rectal damage as a result of childbirth	33 (5.9%)	85 (15.3%)	2.31 (1.12)
8—Suffering a vaginal tear during childbirth	37 (6.6%)	72 (12.9%)	2.12 (1.14)
17—Need a forceps or suction cup	38 (6.8%)	133 (23.9%)	2.48 (1.21)
19—Have an episiotomy performed (have a cut made in your vagina)	82 (14.7%)	76 (13.6%)	1.96 (1.26)
28—Need stitches after childbirth	181 (32.5%)	23 (4.1%)	1.22 (1.12)

M = Mean/SD = standard deviation; ^a^ For each of the nine-subscale listed, sum the items in the subscale. To create mean score (to be able to compare across subscales), divide the subscale score by the number of items in the subscale. ^b^ Only the highest (ceiling) and lowest scores (floor) per question are shown.

**Table 3 jcm-11-01843-t003:** Factorial loads (after rotation) of the 4-factor, 37-item model obtained on the first sample (*n* = 279).

	F1	F2	F3	F4
Item 1	0.097	0.503	0.037	0.101
Item 2	−0.037	0.427	0.276	0.136
Item 3	−0.004	0.659	−0.081	0.089
Item 4	0.442	0.191	−0.136	0.172
Item 5	0.418	0.060	−0.085	0.180
Item 6	0.099	0.872	−0.010	−0.154
Item 7	0.149	−0.177	0.036	0.483
Item 8	−0.037	0.307	0.225	0.259
Item 9	−0.018	0.880	−0.076	0.044
Item 10	0.087	0.886	−0.048	−0.075
Item 12	−0.126	0.139	0.162	0.514
Item 13	−0.158	−0.228	0.014	0.686
Item 15	−0.014	−0.116	−0.026	0.775
Item 16	−0.039	0.882	−0.038	−0.050
Item 17	0.193	0.463	0.109	0.049
Item 18	−0.104	0.362	0.499	0.064
Item 20	0.013	0.817	−0.017	−0.077
Item 21	0.164	−0.142	0.029	0.509
Item 22	0.524	0.074	−0.109	0.131
Item 23	0.323	−0.191	0.043	0.456
Item 24	−0.005	−0.069	0.052	0.812
Item 25	0.314	−0.135	−0.089	0.538
Item 26	−0.002	0.215	−0.076	0.685
Item 27	0.020	0.205	−0.165	0.718
Item 28	0.357	0.001	0.258	0.212
Item 29	−0.075	0.219	0.618	0.004
Item 30	−0.027	0.038	0.800	−0.019
Item 31	−0.087	−0.002	0.471	0.283
Item 32	0.171	−0.100	0.104	0.512
Item 33	0.772	−0.013	−0.061	−0.005
Item 34	0.027	−0.075	0.987	−0.099
Item 35	0.068	−0.075	0.934	−0.050
Item 36	0.882	−0.046	0.042	−0.098
Item 37	0.087	−0.070	0.775	−0.036
Item 38	0.477	0.089	0.289	−0.103
Item 39	0.566	0.224	0.266	−0.172
Item 40	0.634	0.101	0.046	0.015

**Table 4 jcm-11-01843-t004:** Factorial loads (after rotation) of the model obtained from 4-factor, 37-item model in the second sample (*n* = 278) (loadings lower than absolute 0.300 omitted).

	F1 Fear of Medical Interventions	F2 Fear of Harm and Dying	F3 Fear of Pain	F4 Fears Relating to Sexual Aspects and Embarrassment
Item 1—That you are harmed by incompetent medical assistance		0.565		
Item 2—Suffer a tear or rectal damage as a result of childbirth		0.443		
Item 3—Dying during childbirth		0.764		
Item 4—Receive general anaesthesia	0.493			
Item 5—Get injections	0.551			
Item 6—That harm or harm the baby as a result of childbirth		0.894		
Item 7—That other people see you naked during childbirth				0.568
Item 8—Suffering a vaginal tear during childbirth		0.355		
Item 9—That the baby suffers some damage during childbirth		0.940		
Item 10—That harm the baby in a medical intervention during childbirth (e.g., vacuum, anaesthesia, forceps...)		0.890		
Item 12—That your vagina stretches from having a vaginal birth				0.619
Item 13—Enjoy less sexual intercourse by stretching the vagina because of vaginal birth				0.852
Item 15—Make your body less attractive after childbirth				0.688
Item 16—That the baby suffocates during childbirth		0.929		
Item 17—Need a forceps or suction cup		0.474		
Item 18—Not getting the pain medication you need		0.470	0.386	
Item 20—That the baby dies during childbirth		0.715		
Item 21—Other people see you urinating during childbirth				0.565
Item 22—Have your epidural administered	0.703			
Item 23—Feeling observed by strangers during childbirth				0.646
Item 24—Make your vagina look less attractive after a vaginal birth				0.898
Item 25—That you have scars after the caesarean section	0.388			0.376
Item 26—Enjoying sex less because of feeling pain or discomfort after childbirth				0.696
Item 27—That your partner enjoys less of sexual intercourse after childbirth by stretching your vagina after childbirth				0.771
Item 28—Needing stitches after childbirth	0.435			0.309
Item 29—That you do not get the epidural during childbirth in the case of wanting or needing it			0.457	
Item 30—Feeling pain during childbirth			0.846	
Item 31—Having a vaginal birth			0.487	
Item 32—That other people see you defecate during childbirth				0.709
Item 33—Not being able to have a vaginal birth despite being what you prefer	0.758			
Item 34—Feel pain while pushing the baby			0.955	
Item 35—Feeling pain during a vaginal birth			0.962	
Item 36—Have a C-section	0.915			
Item 37—Feeling pain during contractions			0.845	
Item 38—To be probed (a tube that is inserted into the urethra to collect urine	0.534			
Item 39—Feeling pain during a C-section	0.541			
Item 40—Not being able to have the type of birth you would like (for example vaginal or caesarean section)	0.632			

**Table 5 jcm-11-01843-t005:** Correlations (and 95% confidence intervals) between the factors of the obtained model.

Factors	Correlation Values	Bias-Corrected Bootstrap 95% Confidence Intervals
1--------2	0.479 *	(0.403–0.579)
1--------3	0.537 *	(0.480–0.646)
1--------4	0.533 *	(0.464–0.680)
2--------3	0.380 *	(0.288–0.472)
2--------4	0.399 *	(0.315–0.485)
3--------4	0.588 *	(0.539–0.674)

* Significantly different from zero at population.

## Data Availability

The data used in this research are confidential and are protected in a coded and anonymised database kept by the research group in accordance with Spanish regulations. Because of the sensitive nature of the questions asked in this study, respondents were assured that the raw data would be kept confidential and would not be shared. However, the raw data from the CFQ-e survey (response to each item) and without the rest of the sociodemographic–obstetric variables could be shared with those researchers who contact the corresponding authors with a reasonable and logical request.

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
