# Peer review of "Validation and Psychometric Properties of the Spanish Version of the Fear of Childbirth Questionnaire (CFQ-e)"

_jcm, 2022, doi:10.3390/jcm11071843_

Round 1

Reviewer 1 Report

 I have read and reviewed a study which aimed to translate, adapt and validate the Fear of Childbirth Questionnaire (CFQ) for use in Spain, as well as to describe and evaluate its psychometric properties. The findings and conclusions of this study were relevant to the aim addressed by the study. The fear of childbirth has been established as a well known and referred issue but the scientific studies about this issue are limited in number. This study adds to our knowledge about the validation of a questionnaire about childbirth in a cohort of pregnant women and highlights the method for performing in different cohorts with different mother tongues. Since the paper was written clearly and in detail, it was a pleasure reading it and I experienced no comprehensive problem. As for the discussion and conclusion parts, they complied with the findings of the authors and arguments drawn from these findings. 

Reviewer 2 Report

Dear colleagues,

I would like to say thank you for the opportunity to review your manuscript. This is a fundamental work. In addition to fulfilling the STROBE criteria, I want to note the thoughtful style of writing the article. The clearly described practical significance (already in the first paragraph of the introduction), the calculation of the sample size and a detailed description and discussion of statistical analysis show the high level of the authors.

Author Response

Dear Reviewer:

Thank you very much for your kind comments and your time. Your words encourage us to continue working with enthusiasm and will help us to improve in our future research projects.

Best regards,

Reviewer 3 Report

The authors have done an excellent job recruiting a large sample for the validity of this questionnaire to measure a much-needed construct with great clinical applicability. In addition, they have made a great effort to carry out a complete factorial analysis and that all the steps they have carried out are reflected in the manuscript.
However, generally speaking, some parts of the manuscript are very difficult to follow. For example, the results become excessively hard to read and track. Perhaps part of the information and the procedure to reach the final version of the questionnaire can be offered as a supplement for those who want to delve deeper, and leave only the psychometric characteristics of the final questionnaire in the main manuscript, summarizing all of the above. On the other hand, some parts need to review the language to be more direct and simple sentences.
Please read the minor comments below:

“On the other, fear of childbirth ,,” use its abbreviation.

What was the sample size at pretest? This os defined in results section but not in methods. Was sample size calculated?

It his formula ok? : Pa= [N!/(A!( NA)!)] I think exclamations should be removed

How did authors know “How the current pregnancy was achieved” in table 1 if women were pregnant at the moment of the study?

3.2.2. Preliminary Factor Analysis : Neither KMO, RMSEA, Barthel index, etc are defined in methods section. Please define them in statistical methods and add references values for them.

Please revise English to make the document easier to follow. Ie: “Based on this preliminary factor analysis, and as recommended, given the results 325 obtained, a …” this is too long to express a result.

In a total of 557 women completed the questionnaire, why only n=278 were used for the finally CFA analysis?

Why did the sample not fulfilled other questionnaire such as W-DEQ-A to obtain convergent validity?

Why did the authors not performed test-retest reliability?

Author Response

Dear Reviewer, thank you very much for your comments and questions.

A word file is attached, containing all our responses.

With best regards.
